# Radiation Safety Assessment in Prostate Cancer Treatment: A Predictive Approach for I-125 Brachytherapy

**DOI:** 10.3390/cancers16101790

**Published:** 2024-05-07

**Authors:** Ho-Da Chuang, Yu-Hung Lin, Chin-Hsiung Lin, Yuan-Chun Lai, Chin-Hui Wu, Shih-Ming Hsu

**Affiliations:** 1Medical Physics and Radiation Measurements Laboratory, National Yang Ming Chiao Tung University, Taipei 11221, Taiwan; ada6225max@yahoo.com.tw (H.-D.C.); kevain@kfsyscc.org (C.-H.L.); 2Department of Biomedical Imaging and Radiological Sciences, National Yang Ming Chiao Tung University, Taipei 11221, Taiwan; 3Department of Medical Physics, Koo Foundation Sun Yat-sen Cancer Center, Taipei 11259, Taiwan; 4Department of Urology, Koo Foundation Sun Yat-sen Cancer Center, Taipei 11259, Taiwan; linyuhung@kfsyscc.org; 5Department of Radiation Oncology, Changhua Christian Hospital, Changhua 50006, Taiwan; 103424@cch.org.tw; 6Department of Medical Imaging and Radiological Sciences, Central Taiwan University of Science and Technology, Taichung 40601, Taiwan; 7Department of Medical Imaging and Radiological Sciences, Tzu-Chi University of Science and Technology, Hualien 97005, Taiwan

**Keywords:** permanent implantation brachytherapy, Monte Carlo, predictive model, radiation safety assessments

## Abstract

**Simple Summary:**

External dose rate measurements for I-125 permanent radioactive source implantation brachytherapy for prostate cancer requires standardized external dose rate measurements and radiation safety procedures. We developed predictive models for external dose rates to ensure compliance with radiation safety guidelines. We used Monte Carlo simulation and experimental measurements to construct an external dose rate model. The model’s accuracy was validated against external dose rate measurements from clinical patients. This model facilitates the calculation and minimization of family members’ radiation dose, aiding in the development of personalized radiation protection strategies.

**Abstract:**

This study uses Monte Carlo simulation and experimental measurements to develop a predictive model for estimating the external dose rate associated with permanent radioactive source implantation in prostate cancer patients. The objective is to estimate the accuracy of the patient’s external dose rate measurement. First, I-125 radioactive sources were implanted into Mylar window water phantoms to simulate the permanent implantation of these sources in patients. Water phantom experimental measurement was combined with Monte Carlo simulation to develop predictive equations, whose performance was verified against external clinical data. The model’s accuracy in predicting the external dose rate in patients with permanently implanted I-125 radioactive sources was high (R^2^ = 0.999). A comparative analysis of the experimental measurements and the Monte Carlo simulations revealed that the maximum discrepancy between the measured and calculated values for the water phantom was less than 5.00%. The model is practical for radiation safety assessments, enabling the evaluation of radiation exposure risks to individuals around patients with permanently implanted I-125 radioactive sources.

## 1. Introduction

Prostate cancer is the second most common cancer in men worldwide and the fifth most common cause of cancer deaths [1]. There are various treatment options for prostate cancer in men, including permanent implantation (low dose rate, LDR) or temporary brachytherapy (high dose rate, HDR), external beam radiation therapy (EBRT), and radical prostatectomy (RP) [2]. Low-dose-rate brachytherapy can improve cure rates and disease-free survival rates [3,4]. The radioactive isotope I-125 is used for permanent implantation brachytherapy, a common treatment method for localized prostate cancer [5]. Permanent implantation is a brachytherapy treatment for prostate cancer in which I-125 is implanted directly into the prostate. Compared with other methods, it can significantly reduce the dose outside the implantation area, protect organs at risk (rectum, bladder), reduce the incidence of side effects, and allow patients to recover more quickly [6]. After the radioactive source is implanted, the radiation exposure dose received by people close to the patient is most often measured directly. Healthcare facilities are required to manage patient release according to recommendations from the International Commission on Radiological Protection (ICRP), the National Commission on Radiation Protection and Measurements (NCRP), and guidance from the U.S. Nuclear Regulatory Commission (NRC). Patient release guidelines state that the effective dose for comforters and caregivers must not exceed 5 mSv/year, and the effective dose for members of the public must be unlikely to exceed 1 mSv/year [7,8,9,10]. Since the I-125 radioactive source remains in the body permanently, radiation exposure to other people must be as low as reasonably achievable (ALARA). The International Commission on Radiation Units and Measurements (ICRU) defines measurable operational quantities for radiation protection [11,12]. A whole-body exposure dose assessment defines two operational quantities: 1. The ambient dose equivalent H*(10) is used for radiation detection instruments to assess radiation exposure in the workplace and environment. The operational quantity H*(10) of regional radiation monitoring is used to monitor strongly penetrating radiation (energy above 15 keV); 2. The personal dose equivalent Hp(10) is a retrospective assessment of the radiation exposure dose received by an individual person. For permanent implantation of I-125 radioactive sources, radiation detectors are used to detect the radiation dose rate outside the patient’s body. In radiation protection, H*(10) is important in the field of area monitoring since H*(10) allows for the estimation of the effective dose to the human body. This study has two purposes: (1) To verify the accuracy of the measurement and Monte Carlo simulation of the ambient dose rate equivalent H*(10) outside a water phantom with an I-125 radiation source. (2) To develop a set of equation models based on Monte Carlo results to predict the effective dose received by the patient’s family or the public while the radiation source is implanted.

## 2. Materials and Methods

### 2.1. Experimental Measurement

This study used 30 I-125 radioactive sources from the same batch of selectSeed (1 mCi) model 130.002 (Elekta/Nucletron, Veenendaal, The Netherlands). The source structure is shown in Figure 1a. The selectSeed sources were arranged upright and placed within a 5-mm-diameter plastic tandem tube whose thickness was 0.182 mm (Figure 1b). The I-125 radioactive sources were placed in a Mylar window water phantom of 35 × 35 × 37 cm^3^. The Mylar thickness was 0.254 mm, with a window size of 27 × 27 cm^2^. The I-125 radioactive sources were centered in the Mylar window, positioned at distances ranging from 1 cm to 20 cm. When the radioactive source was placed at the set position (1 cm to 20 cm), we measured H*(10) at the water phantom surface, at 30 cm, and at 100 cm. An AT1121 plastic scintillation detector (Atomtex, Minsk, Republic of Belarus) was used to measure the ambient dose equivalent rate H*(10) at the outer surface of the Mylar window water phantom (0 cm) at distances of 30 cm and 100 cm, as shown in Figure 1c. This radiation detection equipment simultaneously displayed the dose rate and the real-time statistical error percentage. The measurement accuracy was recorded when the collected statistical error was below 2.00%. Measurements were taken once a week over a period of nine weeks, yielding a total of nine measurements. The nine measurements were normalized to the measurement with the I-125 source at 1 cm in the water phantom. After subtracting the background dose rate (0.20 μSv/h), the Atomtex AT1121 plastic scintillation detector was used for continuous long-term measurement of radiation ranging from 50 nSv/h to 10 Sv/h, with an energy range of 15 keV–10 MeV.

### 2.2. Monte Carlo Simulation

To assess the accuracy of the model in predicting the patient’s external dose, this study divided the Monte Carlo simulation into two parts: (1) A water phantom was used for measurements, as shown in Figure 1c, with the Monte Carlo simulation geometry exactly matching the actual measurement conditions. We constructed predictive models based on Monte Carlo simulation data and normalized results to the result with the I-125 source at 1 cm in a water phantom in a water phantom. A predictive model was constructed based on Monte Carlo simulation. (2) Twenty-one patients implanted with I-125 sources underwent computed tomography scans of the pelvic region. Using these images, the thickness of the patient’s anterior, posterior, and bilateral body centered on the implanted source was measured. Based on these data, a simple rectangular water phantom model was created to simulate a patient body with a prostate with an implanted I-125 source. Dose calculations were made at various distances from the phantom surface and compared with the measured values. Both parts of the simulation calculations were performed using the Particle and Heavy Ion Transport Code System (PHITS). In the PHITS code [13], the calculation of the ambient dose equivalent is based on the fluence-to-ambient-dose equivalent conversion coefficients reported in ICRP 74 [14]. To ensure that the statistical error was less than 3%, the number of particles simulated each time was at least 10^9^.

### 2.3. Patient Information and IRB

From May 2013 to February 2016, 21 patients underwent I-125 permanent implant brachytherapy. The prostate was implanted by either brachytherapy alone or brachytherapy combined with external radiation therapy. The prescribed doses of interstitial LDR prostate I-125 brachytherapy were either as monotherapy with a minimum peripheral dose (MPD) of 145 Gy or in combination with external irradiation with an MPD of 110 Gy [15]. Seventeen patients received monotherapy, and 4 patients received combined external beam radiation therapy at KFSYSCC. According to intraoperative planning dosimetry, the isodose 90% of the prostate volume (prostate D90) was 100–130% of the prescribed dose. After the post-implant computed tomography treatment plan was implemented, the day 0 post-implant dosimetry was evaluated, and the patient’s day 0 post-implant computed tomography images were measured from the center of the prostate to the anterior, posterior, left, and right. Water equivalent thickness (WET) was measured in different directions. The sum of the implanted radioactive source activity and the water equivalent thickness (WET) was recorded from post-implant computed tomography images in four different measurement directions (anterior, posterior, left, and right). The WET and external radiation dose rate (μSv/h) were measured in four different directions and at various distances from the patient’s body. This study received ethical approval from the Health Medical Research Ethics Committee of the National Health Research Institutes (NHRI), approval number EC1020101-F.

### 2.4. Clinical Patient External Data Validation

In this study, 21 patients were implanted with I-125 radioactive sources. The average total implant activity of the implanted seeds was 35.15 ± 8.73 U. WETs were measured in four different directions (anterior, posterior, left, and right) from the prostate center to the patient’s body surface using day 0 post-implant computed tomography images. The distance of the implant I-125 radioactive sources center to body surface was 10.33 ± 1.42 cm for each patient in the anterior WET direction, 10.41 ± 0.92 cm in the posterior WET direction, 17.19 ± 1.51 cm in the left WET direction, and 17.24 ± 1.53 cm in the right WET direction. The WETs in the four directions from the prostate center to the body surface were input into the predictive model. Calculations were performed at the body surface (0 cm), 30 cm, and 100 cm H*(10) and compared with the actual external measurements of patients [16,17,18,19,20].

### 2.5. Statistical Analysis

This study used a nonparametric test (Mann–Whitney U test, two-tailed) to compare the differences between experimental measurements and Monte Carlo simulation groups. A value of *p* < 0.05 was considered statistically significant. Statistical analyses were performed with IBM SPSS Statistics (v21, IBM, Armonk, NY, USA). This study used clinical data to validate the predictive model.

## 3. Results

The results of the Monte Carlo simulation of the external dose distribution of the I-125 at different depths in the Mylar window water phantom are shown in Figure 2. The ambient dose equivalent H*(10) at a certain distance from the phantom surface decreased exponentially with increasing depth of the I-125 sources. At the three distances from the phantom surface (0, 30, and 100 cm), the exponential decay of H*(10) was derived as a function of the equivalent water thickness z from the center of the radiation source to the water phantom surface. A total of 1–3 exponential equation functions were constructed, each with R^2^ > 0.999, as follows:Normalized H*(10) _0 cm_ = 1.7811 e ^−0.5841z^(1)
Normalized H*(10) _30 cm_ = 1.4360 e ^−0.3555z^(2)
Normalized H*(10) _100 cm_ = 1.4026 e ^−0.3273z^(3)

All the measured values were normalized as described in Section 2.2, and the H*(10) and Monte Carlo simulation values were compared. The measurement results are shown in Figure 3. The average difference at the water phantom surface (0 cm) was 3.14 ± 0.44%, the average difference at 30 cm from the water phantom was 1.92 ± 0.68%, and the average difference at 100 cm from the water phantom was 2.48 ± 0.76%. For the water phantom surface, the average difference between the measured and Monte Carlo simulation results was 3.30 ± 0.16%. At a distance of 30 cm from the water phantom, the average difference between the measured and Monte Carlo simulations was 3.74 ± 1.26%. At a distance of 100 cm from the water phantom, the average difference between the measured and Monte Carlo simulations was 4.04 ± 1.65%. Overall, the differences between the measured and Monte Carlo simulations were within 5.00%. According to the ICRP 98 report [7], the external dose around the patient with a permanent implant depends on (1) the total number and total activity of the radioactive seeds implanted in the patient; (2) the geometric distribution of seeds in the prostate; and (3) the attenuation of radiation caused by the thickness and composition of the patient’s tissue [7]. Therefore, the external radiation dose rate is positively correlated with the sum of the source activities of the implanted radioactive nuclei and negatively correlated with the implantation depth of the radioactive source. In the predictive model, we must consider the initial dose rate of I-125 implantation [initial dose rate (D_0_) in water = S_k_·Λ] [16,17,18,19,20] and the air-kerma strength and H*(10) conversion factor [21,22,23,24,25,26]. The equations for this are as follows:H*(10) _0 cm_ = S_k_·Λ·CF·1.7811 e ^−0.5841z^(4)
H*(10) _30 cm_ = S_k_·Λ·CF·1.4360 e ^−0.3555z^(5)
H*(10) _100 cm_ = S_k_·Λ·CF·1.4026 e ^−0.3273z^(6)
where S_k_ is the total air-kerma strength of all sources implanted; Λ is the dose rate constant, which is the dose rate of the unit air-kerma strength source on the horizontal axis 1 cm away from the water in the water phantom; and CF is the conversion factor that converts air-kerma strength into ambient dose rate equivalent H*(10) (Sv/Gy) [21,22,23,24,25,26]. The average energy of the I-125 source used in this study was 27.4 keV, and the dose rate conversion factor for the external photon air exposure dose rate was calculated from the ICRP-74 (ICRP, 1997) [14]. According to ICRP Report No. 74, the conversion factor from air kerma to the ambient dose rate equivalent H*(10) is approximately equal to 1.00 [14].

The H*(10) values measured in four directions at 0 cm, 30 cm, and 100 cm from the body surface were similar to the predicted values (Mann–Whitney U test) (Figure 4).

## 4. Discussion

The ambient dose equivalent H*(10) plays a critical role in area dose measurement and radiation protection. H*(10) represents the radiation exposure dose determined through environmental area measurements when assessing external radiation in patients with implanted sources [7,8,9,10,11]. In this study, to ensure accuracy and consistency, repeated measurements were conducted, and the measurement statistical error was reduced to less than 2.00%. When comparing the measured and calculated values, the dose rate deviations at each location were less than 5.00%. According to the experimental measurements, when the I-125 source and survey meter were closer to the body surface, the predicted–measured difference would increase. This is mainly affected by the different media of the water phantom and air. The difference diminishes when the I-125 sources and survey meter are further from the body surface. This is mainly affected by the inverse square law. In validating the prediction model based on clinical measurement data, the calculation of H*(10) in the anterior and posterior directions of the patient’s body surface (0 cm) showed a large difference in H*(10). The reason is that the prostate has a certain geometric volume, and there is a minor error in WET measurement (1–2 mm). The measured H*(10) is therefore distributed over a wide range and is significantly lower than the predictive model value by approximately 10%. In the calculation of H*(10) in the left and right directions of the patient’s body surface (0 cm), the difference in H*(10) is small. When the radiation source implant depth is greater than 15 cm, the difference in H*(10) is less than 5%. At radiation source depths greater than 10 cm and distances of 30 cm and 100 cm, the H*(10) difference was less than 5%. When using the predictive model to calculate the H*(10) in the external exposure dose rate of public people around a patient, the equivalent water thickness in the permanently implanted patient’s body and the distance outside the body, as well as the sum and attenuation of the radioactive source activity, must be considered. In practice, at distances greater than three times the size of the prostate, the ambient exposure dose rate H*(10) can be considered equal to the geometric center of gravity located at the implant, whose activity is equal to the total activity of the implant. The half-value layer of radiation emitted by I-125 seeds is approximately 2 cm WET for practical radiation protection purposes. This ambient exposure dose rate H*(10) value can be estimated based on the inverse square law and WET attenuation. Therefore, the H*(10) deviation will be reduced at 30 cm and 100 cm [15]. The available data show that in most cases, the dose to the patient’s comforters and carers remains well below the recommended limit of 1 mSv/year. Some members of the household cannot be considered comforters or caregivers. This is the case for pregnant women since the dose to an unborn child should be kept below 1 mSv throughout the pregnancy. Only pregnant women and babies at the time after implantation may need specific precautions. This prediction model facilitates the calculation and minimization of families’ radiation doses, aiding in the development of family radiation protection and precautionary strategies.

### Limitations

There are limitations to this study. First, to verify the strength of the thirty selectSeedsources, the air-kerma strength of the implanted radioactive source was measured. The stated accuracy of the well chamber calibration was ±1.5%, and the measurements of the same thirty sources were repeated. The strength of the thirty radioactive sources of air kerma was 2.18 ± 2.00%. This overall difference is considered acceptable under the ±3.0% threshold set by AAPM TG-56 [27].

Second, an I-125 radioactive source (130.002) was selected for our study. According to the radioactive source activity calibration provided by the manufacturer, there is an uncertainty of 4% [28,29,30,31,32]. The dose rate constant Λ used in the patient treatment planning system was 0.954 cGy·h^−1^·U^−1^, as reported by Karaiskos, while the dose rate constant was _TLD_Λ = 0.938 cGy·h^−1^·U^−1^, as reported by Anagnostopoulos. Nath recommends using consensus based on weighted averages of _MONTE_Λ and _TLD_Λ. For selectSeed, the _CONSENSUS_Λ was 0.946 cGy·h^−1^·U^−1^, which is 0.80% lower than that currently used. The reported uncertainty is 8.00% (Monte Carlo-derived Λ: 0.50%; TLD-derived Λ: 6.90%; and source intensity check: 4.00%) [28,29,30,31,32]. The patient’s external radiation dose was measured using the radiation detector Atomtex AT1121, which was calibrated by the Atomic Science and Technology Development Center of National Tsinghua University. The reported uncertainty is 5.00%.

Third, the computed tomography scan field of view was 65 cm in 512 × 512 pixels, so the pixel size was 1.27 mm × 1.27 mm. The prostate center was defined by a 3D drawing of prostate contouring in maximum transverse geometry and longitudinal geometry boundary edge. Based on the patient’s day 0 post-implant computed tomography imaging data, there was an error of 1–2 mm in the WET from the prostate center to the body surface in the four different measurement directions.

Finally, only 21 patients had clinical data available to verify the feasibility of the predictive model, a relatively small sample.

## 5. Conclusions

This study’s predictive model exhibited a prediction uncertainty of less than 10% for the external dose rate around patients with permanently implanted I-125 radioactive sources. Our findings will be useful for radiation safety assessment in assessing the risk of radiation exposure to patients’ families or the public around patients who have permanently implanted I-125 radioactive sources. The low acceptable radiation dose to the public can be calculated in advance, and a personalized radiation protection baseline strategy can be devised. This can effectively guide radiation protection actions for prostate brachytherapy patients so that people who come into contact with the patient can be exposed to radiation that meets the ALARA standard.

## Figures and Tables

**Figure 1 cancers-16-01790-f001:**
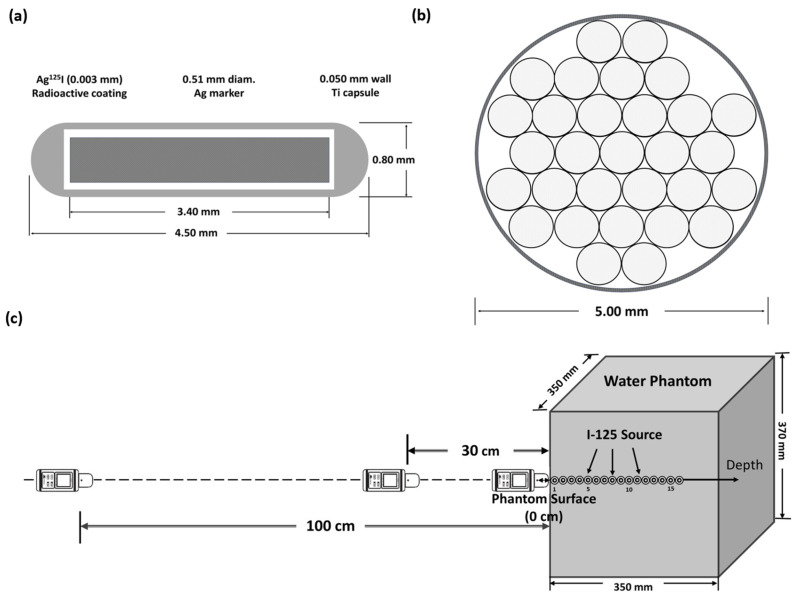
Geometric model of the Monte Carlo simulation and experimental design. (**a**) Schematic diagram of the Nucletron selectSeed I-125, (**b**) 30 Nucletron selectSeed I-125 sources in the plastic tube, and (**c**) the experimental design front view of the ambient dose rate measurement.

**Figure 2 cancers-16-01790-f002:**
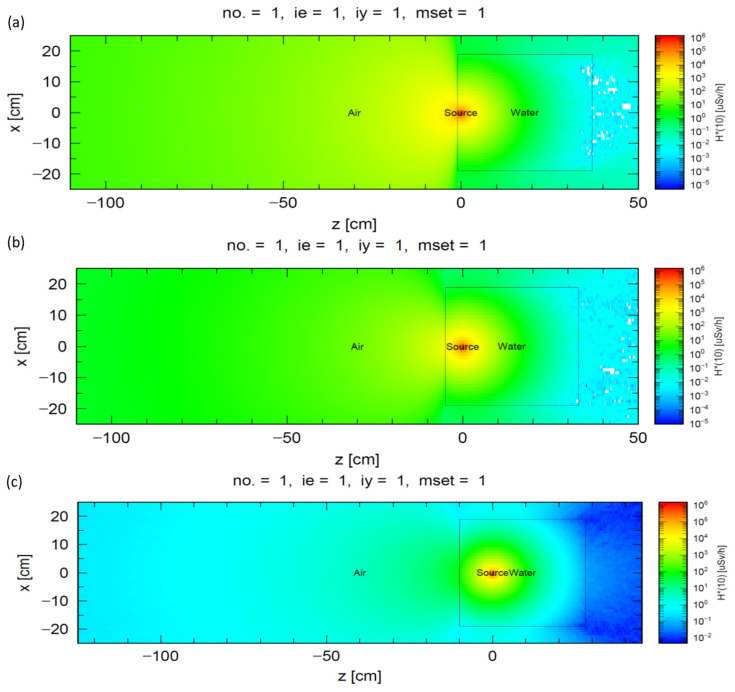
The ambient dose equivalent according to Monte Carlo simulation in the horizontal plane through the I-125 sources at depths of 1 cm (**a**), 5 cm (**b**), and 10 cm (**c**) in water phantom.

**Figure 3 cancers-16-01790-f003:**
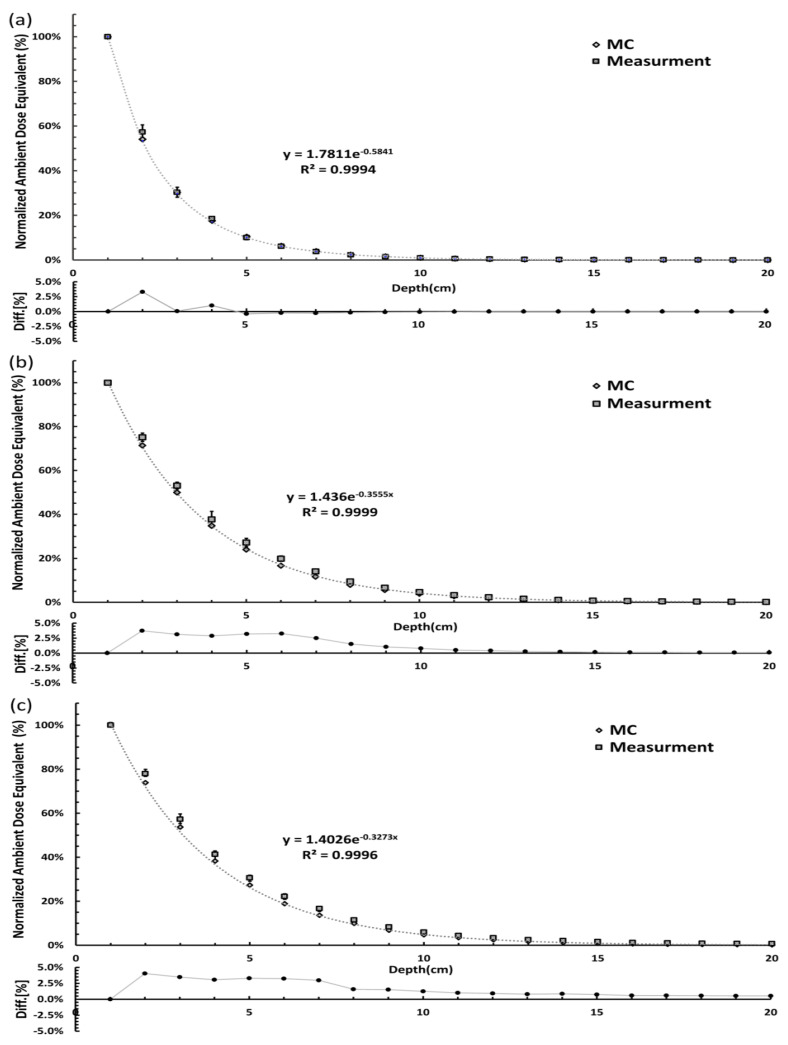
Comparison of the measured (square) and Monte Carlo simulation (rhombus) normalized ambient dose equivalent curves at the phantom surface (0 cm) (**a**) 30 cm (**b**) 100 cm (**c**). The ambient dose equivalent is normalized to the result with the I-125 source at 1 cm from the phantom surface. The percentage ambient dose differences between the simulation and measurements are plotted in the lower panels.

**Figure 4 cancers-16-01790-f004:**
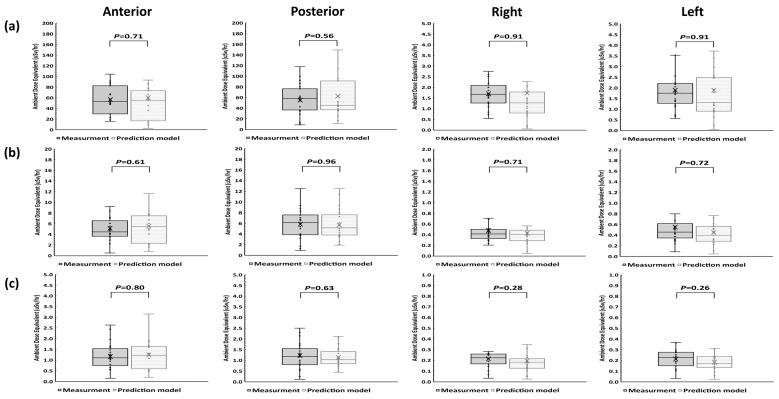
Comparison of measurements and predictive model calculations at the body surface (**a**), at 30 cm (**b**), and 100 cm (**c**), in 4 different directions using the Mann–Whitney U test.

## Data Availability

All data supporting the findings of this study are available within the article, and from the corresponding author(s) upon reasonable request.

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
