# Peer review of "Radiation Safety Assessment in Prostate Cancer Treatment: A Predictive Approach for I-125 Brachytherapy"

_cancers, 2024, doi:10.3390/cancers16101790_

Round 1
Reviewer 1 Report
Comments and Suggestions for Authors
see uploaded PDF with comments

Comments on the Quality of English LanguageEnglish language must be improved; some phrases were hard to read for reviewer; in the uploaded document I give many examples of sentences that should be rephrased.
Author Response
Response to reviewer 1 comments
I would like to extend my sincere gratitude for the time and effort you have dedicated to reviewing our manuscript. Please find the detailed responses below and the corresponding revisions in the re-submitted files.
Point-by-point response to comments and suggestions for authors
Comments 1. dose rate measurements
Response 1: Thank you for your valuable feedback. We revise the statement. [P.1, line 22]
Comments 2. I suppose the accuracy of the predictive model?
Response 2: Thank you for your valuable feedback. The accuracy is for the predictive model. [P.1, line 27]
Comments 3. was high with R2=0.999
Response 3: Thank you for your valuable feedback. We revise the statement. [P.1, line 33]
Comments 4. consider to add reference Henry et al. 2022. GEC-ESTRO ACROP prostate brachytherapy guidelines Radiotherapy and Oncology, 167, 244-251
Response 4: Thank you for your valuable feedback. We add the reference [6] Henry et al. 2022. GEC-ESTRO ACROP prostate brachytherapy guidelines Radiotherapy and Oncology, 167, 244-251 [P.2, line 51]
Comments 5. “any other individual” or “radiological worker”?
Response 5: Thank you for your valuable feedback. We revise the “any other individual” to “comforters and caregivers”. The radiation protection is about the family members. [P.2, line 61]
Comments 6. Rephrase this sentence; two operational quantities, but only one quantity is mentioned (in this sentence).
Response 6: Thank you for your valuable feedback. We revise the two operational quantities H*(10) and Hp(10) sentence. [P.2, line 67-76]
Comments 7. Rephrase: H*(10) is not a volume
Response 7: Thank you for your valuable feedback. We revised to the operational quantity H*(10). [P.2, line 69-70]
Comments 8. Or “received by an individual person”/
Response 8: Thank you for your valuable feedback. We revised the “ individual Personal dosage” to “received by an individual person”. [P.2, line 72]
Comments 9. I would suggest (1) To verify the accuracy of measurement and Monte Carlo simulation of ambient dose rate equivalent H*(10) outside a water phantom with a I-125 radiation source, (2) To develop ...
Response 9: Thank you for your valuable feedback. We revise the “To verify the accuracy of ambient dose rate equivalent 76 H*(10) experimental measurement and Monte Carlo Simulation of I-125 radiation source 77 outside the water phantom. (” to “To verify the accuracy of measurement and Monte Carlo simulation of ambient dose rate equivalent H*(10) outside a water phantom with a I-125 radiation source,”. [P.2, line 76-78]
Comments 10. please give some more detail regarding accuracy of Monte Carlo calculations. And refer to benchmark calculations e.g.
Response 10: Thank you for your valuable feedback. To ensure that the statistical error was less than 3%, the number of particles simulated each time was at least 109. In this study, both the Monte Carlo calculated values and the measured values were normalized. After normalization, the maximum discrepancy between the calculated and measured values was approximately 5%, as shown in Figure 3.
Comments 11. please more detail, e.g. cylindrical rod (diameter and length), rod is coated with a silver halide layer (AgCl/AgI), titanium encapsulation (diameter and thickness)
Response 11: Thank you for your valuable feedback. The Figure 1(a) We revised the source component.
Comments 12. I count 31 sources in figure 1(b)?
Response 12: Thank you for your valuable feedback. The Figure 1(b) We revised the number of sources.
Comments 13. I do not understand how sources are arranged in the phantom
Response 13: Thank you for your valuable feedback. When the radioactive source is at 1 cm away from the water phantom, we measure H*(10) on the outer surface of the water phantom, 30 cm and 100cm. As the depth of the radioactive source increases in the water phantom, we repeatedly measure the outside of the water phantom of three positions. The nine-measurement data all normalized at I-125 sources at 1 cm in water phantom. [P.3, line 99]
Comments 14. replace "measurement experiments" by "measurements"
Response 14: Thank you for your valuable feedback. We revised the “measurement experiments” to “measurements”. [P.3, line 97]
Comments 15. according to the instrument specifications, the intrinsic relative measurement error is +/-15% max (for continuous long-term and short-term radiation) in the energy range 15keV to 10 MeV.
I do not understanfd the 0.05 microSv/h (after subracting background)
Response 15: Thank you for your valuable feedback. We revised the “The measurement accuracy was recorded when the collected statistical error was below 2.00%. Measurements were taken once a week over a period of nine weeks, yielding a total of nine measurements. The nine measurements were all normalized at 1 cm from the I-125 source in the water phantom. After subtracting the background dose rate (0.20 μSv/h), the Atomtex AT1121 plastic scintillation detector was used for continuous long-term radiation ranging from 50 nSv/h to 10 Sv/h, with an energy range of 15 keV – 10 MeV.” [P.3, line 96-102]
Comments 16. 31?, see figure
Response 16: Thank you for your valuable feedback. The Figure 1(b) We revised the number of sources.
Comments 17. I do not understand figure c; are the I-125 sources positioned in a row in the phantom? or put together in a tube at a certain depth?
Response 17: Thank you for your valuable feedback. We put the 30 sources together in a tube and the I-125 sources changed position (1 cm, 2cm, to 20 cm) inside the water phantom. The nine-measurement data normalized at I-125 sources at 1 cm in water phantom.
Comments 18. replace "The sum of ....from the patient's body" to an other section, e.g. Experimental data
Response 18: Thank you for your valuable feedback. We replaced "The sum of ....from the patient's body" to an other section, e.g. Experimental data
Comments 19. post-live implant? I suggest "post-implant"
Response 19: Thank you for your valuable feedback. We revised the “post-live implant” to “post-implant”. [P.4, line 134-151]
Comments 20. please rephrase: The WET can not be measured at various distances from the patient's body
Response 20: Thank you for your valuable feedback. Water equivalent thickness (WET) was measured in different measurement directions. The sum of the implanted radioactive source activity and the water equivalent thickness (WET) was recorded from post-implant computed tomography images in four different measurement directions (anterior, posterior, left, and right). The WET and external radiation dose rate (μSv/h) were measured in four different directions and at various distances from the patient’s body. [P.4, line 137-142]
Comments 21. suggestion to rephrase: "To assess the accuracy of the model to predict the patient's external dose, this ..."
Response 21: Thank you for your valuable feedback. To assess the accuracy of the model in predicting the patient's external dose, this study divided the Monte Carlo simulation into two parts: (1) A water phantom was used for measurements, as shown in Fig. 1(c), with the Monte Carlo simulation geometry ex-actly matching the actual measurement conditions. We constructed predictive models based on Monte Carlo simulation data and normalized I-125 sources at 1 cm in a water phantom. A predictive model was constructed based on Monte Carlo simulation. [P.3, line 108-113]
Comments 22. please rewrite section 2.3; If I understand correctly, dimensions were measured in CT images of 21 patients and based on this an average phantom was chosen for Monte Carlo Simulation and measurements. I would not call that Monte Carlo Simulation in two parts; may be confusing
Response 22: Thank you for your valuable feedback. We rewrite section 2.2, 2.3 and 2.4; first Experimental measurement--> Monte Carlo simulation--> Build predictive model--> Predictive model statistical test-->External clinical verification. [P.3-4, line 107-166]
Comments 23. not clear; do authors mean "the distance from the center of the sources to the patient's body surface in anterior, posterior, left and right direction"?
Response 23: Thank you for your valuable feedback. We rewrite the “To assess the accuracy of the model in predicting the patient's external dose, this study divided the Monte Carlo simulation into two parts: (1) A water phantom was used for measurements, as shown in Fig. 1(c), with the Monte Carlo simulation geometry ex-actly matching the actual measurement conditions. We constructed predictive models based on Monte Carlo simulation data and normalized I-125 sources at 1 cm in a water phantom. A predictive model was constructed based on Monte Carlo simulation. (2) Twenty-one patients implanted with I-125 sources underwent computed tomography scans of the pelvic region. Using these images, the thickness of the patient's anterior, posterior, and bilateral body centered on the implanted source was measured. Based on these data, a simple rectangular water phantom model was created to simulate a patient body with a prostate with an implanted I-125 source. Dose calculations were made at various distances from the phantom surface and compared with the measured values.” [P.3, line 108-119]
Comments 24. a phantom based on average patient dimensions or phantoms for each individual patient?
Response 24: Thank you for your valuable feedback. A water phantom measurement data for each individual patient.
Comments 25. suggestion: "to simulate a patient body with prostate and I-125 sources". (if I understand correctly)
Response 25: Thank you for your valuable feedback. We rewrite " Based on these data, a simple rectangular water phantom model was created to simulate a patient body with a prostate with an implanted I-125 source. ". [P.3, line 116-118]
Comments 26. PHITS code references?
Response 26: Thank you for your valuable feedback. The PHITS code references:
[13]. Sato, T., Niita, K., Matsuda, N., Hashimoto, S., Iwamoto, Y., Noda, S., ... & Sihver, L. (2014). Overview of the PHITS code and its application to medical physics. Prog. Nucl. Sci. Technol, 4, 879-882. [P.3, line 119-120]
Comments 27. title of this section? maybe change into "Experimental data"
Response 27: Thank you for your valuable feedback. We rewrite it into "Experimental data".
Comments 28. rephrase: In this study, 21 patients were implanted with I-125 radioactive sources. Average (range) of total air kerma strength of implanted seeds amounted xx (xx-xx) cGy/h cm2 (or other comparable unit).
Response 29: Thank you for your valuable feedback. We rewrite “In this study, 21 patients were implanted with I-125 radioactive sources. The average air kerma strength of implanted seeds amounted 35.15±8.73 U.” [P.4, line 145-146]
Comments 30. which "predictive model"? there is no mention of a predictive model in the Materials and Methods section so far. Only shortly mentioned in the Introduction section
Response 30: Thank you for your valuable feedback. We base on Mote Carlo simulation to construct predictive model. [P.3, line 113]
Comments 31. replace "under the Mylar window" by "in the"
Response 31: Thank you for your valuable feedback. We replace "under the Mylar window" by "in the".[P.4, line 169]
Comments 32. Ambient dose equivalent H*(10) at a certain distance from the phantom surface proved to decay exponentially with increasing depth of the I-125 sources,
Response 32: Thank you for your valuable feedback. We rewrite Ambient dose equivalent H*(10) at a certain distance from the phantom surface proved to exponential attenuation with increasing depth of the I-125 sources. [P.4-5, line 169-171]
Comments 33. At 3 distances from the phantom surface (at 0, 30 and 100 cm), the exponential decay of H*(10) was derived as a function of the equivalent water thickness z from the center of the radiation sources to the water phantom surface. How was H*(10) normalized? Equations 1-3 fit the Monte Carlo data well with R2>0.999.
Response 33: Thank you for your valuable feedback. In section 2.1 Experimental measurement and 2.2 Monte Carlo Simulation, we write the “The nine-measurement data all normalized at I-125 sources at 1 cm in water phantom” and “We are constructing predictive models based on Monte Carlo simulation data and normalized at I-125 sources at 1 cm in water phantom”. [P.3, line 99, 111, 112]
Comments 34. "x" is "z" according to figure 2?
Response 34: Thank you for your valuable feedback. The "x" is "z", we rewrite x is z. [P.5-6, line 174, 178, 179, 180, 208, 209, 210]
Comments 35. The ambient dose equivalent according to Monte Carlo simulation in the horizontal plane through the I-125 source at 1, 5, 10 cm depth, respectively.
Are the values in figure 2 normalized?
Response 35: Thank you for your question regarding the values presented in Figure 2. The ambient dose equivalent values depicted at 1, 5, and 10 cm depths from the I-125 source are absolute values from the Monte Carlo simulations and are not normalized.
Comments 36. The measured H*(10) values were normalized to ...... and compared to the Monte Carlo simulation results (see Figure 3)
Response 36: Thank you for your valuable feedback. The measured H*(10) values were normalized to ...... and compared to the Monte Carlo simulation results.
Comments 37. What do the authors mean here? difference between measurement and Monte Carlo simulation?
Response 37: Thank you for your valuable feedback. We rewrite that mean±SD. At the same time, the experimental measurement values were normalized in the same way, and the H*(10) and Monte Carlo simulation values were compared. The re-sults are shown in Figure 3. In experimental measurement, the average differences at the water phantom surface (0 cm) were 3.14±0.44%. The average differences at a distance of 30 cm from the water phantom are 1.92±0.68%. The average differences at a distance of 100 cm from the water phantom are 2.48±0.76%. In the water phantom surface, the aver-age differences between the experimental measured and Monte Carlo simulation are 3.30±0.16%. At a distance of 30 cm from the water phantom, the average differences be-tween the experimental measured and Monte Carlo simulation are 3.74±1.26%. At a distance of 100 cm from the water phantom, the average differences between the experimental measured and Monte Carlo simulation are 4.04±1.65%. [P.5-6, line 186-196 ]
Comments 38. how was difference calculated? as a percentage of the normalized value or as a percentage of the local value? or otherwise, please explain. and how was average difference calculated? over a certain range?
Response 38: Thank you for your valuable feedback. We rewrite that mean±SD. At the same time, the experimental measurement values were normalized in the same way, and the H*(10) and Monte Carlo simulation values were compared. The results are shown in Figure 3. In experimental measurement, the average differences at the water phantom surface (0 cm) were 3.14±0.44%. The average differences at a distance of 30 cm from the water phantom are 1.92±0.68%. The average differences at a distance of 100 cm from the water phantom are 2.48±0.76%. In the water phantom surface, the aver-age differences between the experimental measured and Monte Carlo simulation are 3.30±0.16%. At a distance of 30 cm from the water phantom, the average differences be-tween the experimental measured and Monte Carlo simulation are 3.74±1.26%. At a distance of 100 cm from the water phantom, the average differences between the experimental measured and Monte Carlo simulation are 4.04±1.65%. [P.5-6, line 186-196 ]
Comments 39. Again differences? How are these different from the differences mentioned in the sentences above?
Response 39: Thank you for your valuable feedback. We rewrite that mean±SD. At the same time, the experimental measurement values were normalized in the same way, and the H*(10) and Monte Carlo simulation values were compared. The re-sults are shown in Figure 3. In experimental measurement, the average differences at the water phantom surface (0 cm) were 3.14±0.44%. The average differences at a distance of 30 cm from the water phantom are 1.92±0.68%. The average differences at a distance of 100 cm from the water phantom are 2.48±0.76%. In the water phantom surface, the aver-age differences between the experimental measured and Monte Carlo simulation are 3.30±0.16%. At a distance of 30 cm from the water phantom, the average differences be-tween the experimental measured and Monte Carlo simulation are 3.74±1.26%. At a distance of 100 cm from the water phantom, the average differences between the experimental measured and Monte Carlo simulation are 4.04±1.65%. [P.5-6, line 186-196 ]
Comments 40. calculated with predictive model or Monte Carlo simulation? please be clear; not clear to reviewer.
Response 40: Thank you for your valuable feedback. The predictive model that we refer to in our manuscript is also based on the results derived from these Monte Carlo simulations. This model uses the simulation data to predict H*(10) under various conditions and configurations.
Comments 41. rephrase this sentence; not clear; e.g. an equation is not emitted; to explain attenuation and scattering?
Response 41: Thank you for your valuable feedback. We rewrite this sentence about the attenuation and scattering. [P.6, line 198-201]
Comments 42. there is no term time (t) in the equations; I do agree, that we have to look at initial H*(10) at the time the patient leaves the hospital
Response 42: Thank you for your valuable feedback.
Comments 43. is [11] a relevant reference for initial dose rate?
Response 43: Thank you for your valuable feedback. We change the relevant reference for initial dose rate. [P.6, line 204-205]
Comments 44. Is a ROC analysis suitable for the comparison between measurements and predictive model? What are in this case true positive, false positive, true negative, false negative outcomes?
I think the comparison in figure 4 is enough, the ROC analysis and figure 5 may be omitted.
Response 44: Thank you for your valuable feedback. We add the sensitivity, specificity, Positive predictive value (PPV) and Negative predictive value (NPV).The AUC results at the patient's surface, 30 cm, and 100 cm, and in four directions, over-all, the AUC are all greater than 0.759. The AUC at the Body surface is 0.759-0.833, sensi-tivity is 0.612-0.911, specificity is 0.602-0.897, Positive predictive value (PPV) is 0.766-0.852, Negative predictive value (NPV) is 0.789-0.834. The AUC at 30 cm is 0.870-0.891, sensitivity is 0.822-0.953, specificity is 0.817-0.937, Positive predictive value (PPV) is 0.866-0.952, Negative predictive value (NPV) is 0.859-0.934. The AUC at 100 cm is 0.912-0.981, sensitivity is 0.901-0.991, specificity is 0.881-0.988, Positive predictive value (PPV) is 0.896-0.982, Negative predictive value (NPV) is 0.898-0.984. [P.6, line 221-227]
Comments 45. in discussion, please elaborate on why this predictive model is relevant for clinical practice. will safety measures for patient's family or others depend largely on the outcome of the predictive model?
Response 45: Thank you for your valuable feedback. The available data show that, in most cases, the dose to comforters and carers remains well below the recommended limit of 1 mSv/year. Some members of the household can-not be considered comforters and caregivers. This is the case for pregnant women since the dose to an unborn child should be kept below 1 mGy over the term of the pregnancy. Only the case where the pregnant woman and children at the time of post-implantation may need specific precautions. This predicted model facilitates the calculation and minimization of the families' radiation dose, aiding in the development of family radiation protection and precaution strategies. [P.10, line 281-288]
Comments 46. The H*(10) represents the radiation exposure dose determined through environmental area measurements when assessing external radiation in patients with implanted sources [5,10].
Response 46: Thank you for your valuable feedback. We rewrite the statement.
Comments 47. Or is it just inverse square law; how were relative differences calculated; normalization?
Response 47: Thank you for your valuable feedback. When the I-125 sources implant in short distance (< 7 cm, high gradient area in Figure 3.) in patients and the measurements near the patients’ body surface, therefore the external dose rate variation is large. If the distance is large enough from patient, we can roughly calculation by inverse square law (by ICRP 98).
Comments 48. smaller depths of sources?
Response 48: Thank you for your valuable feedback. We rewrite the statement “In verifying the predictive model based on clinical measurement data, the H*(10) at the body surface in the patient's anterior and posterior directions, especially for short depths because in the predicted model the radiation sources in high gradient depth (<10 cm), so the measured H*(10) is distributed over a wide range in high gradient depth. Within and is significantly lower than the predicted model values by approximately 10%.” [P.6, line 259-264]
Comments 49. please rephrase; it is not the deviation of the survey meter; I suppose it is deviation between measurement and model?
Response 49: Thank you for your valuable feedback. We rewrite “Within and is significantly lower than the predicted model values by approximately 10%. When the radiation sources depth over 10 cm are at 30 cm and 100 cm, its H*(10) difference is less than 5%. When based on the predicted model to calculate the H*(10) in external exposure dose rate of public people around a patient, the equivalent water thickness in the permanently implanted patient's body and the distance outside the body, as well as the sum and attenuation of the radioactive source activity, must be considered. When the patient's body thickness is more than three times the size of the prostate, the actual due to distribution of sources can be considered equal to the H*(10) due to a single with strength equal to the total strength of individual sources.” [P.6, line 264-271]
Comments 50. effective dose in dose rate? what do authors mean? please rephrase;
Response 50: Thank you for your valuable feedback. We rewrite the “exposure dose rate”.
Comments 51. not clear; in sentence above "considered equal"
Response 51: Thank you for your valuable feedback. We rewrite the “when the I-125 radiation sources are distributed in the prostate under the size and geometric shap and the distance is more than three times, the H*(10) deviation will be reduced at 30 cm and 100 cm”. [P.6, line 272-274]
Comments 52. why not use the consensus data in planning system? and furthermore: was the treatment planning system used in this analysis? and if so, how?
Response 52: Thank you for your valuable feedback. Excellent agreement between TLDΛ and CONΛ was observed for all source models that currently have an AAPM recommended consensus dose-rate constant value. These results demonstrate that the PST is an accurate and robust technique for the determination of dose rate constant for low-energy brachytherapy sources. By references [38.] Chen, Z. J., & Nath, R. (2010). A systematic evaluation of the dose-rate constant determined by photon spectrometry for 21 different models of low-energy photon-emitting brachytherapy sources. Physics in medicine and biology, 55(20), 6089–6104.
Comments 53. error may be larger? how was the prostate center determined?
Response 53: Thank you for your valuable feedback. the computed tomography scan field of view is 65 cm in 512x512 pixels, so the pixel size is 1.27 mm x1.27 mm. The prostate center is defined by a 3D drawing of prostate con-touring in maximum transversal geometry and longitudinal geometry boundary edge. [P.11, line 307-309]

Reviewer 2 Report
Comments and Suggestions for Authors
This study was reported the safety assessment of brachytherapy for prostate cancer. The reviewer would like to suggest some critiques as follows.
1. On line 22, what is “public’s radiation dose”? The reviewer thinks the wording is too rough to be included in the summary.
2. On line 53, during BT, the seed is inserted into the prostate, not directly into the tumor. It may be difficult to identify the exact lesion on imaging.
3. On line 53, a citation regarding this statement should be provided.
4. On line 54, what is “normal tissue”? surrounding tissues of the prostate?
5. Limitation is described incorrectly.
6. Discussion is too short if it is only a discussion of the results of the experiment.
7. The authors should consider how it affects the actual treatment, how effective the treatment is, and how to minimize the impact on the surroundings and maximize the therapeutic effect.
Author Response
Response to reviewer 2 comments
I would like to extend my sincere gratitude for the time and effort you have dedicated to reviewing our manuscript. Please find the detailed responses below and the corresponding revisions in the re-submitted files.
Point-by-point response to comments and suggestions for authors
Comments 1. On line 22, what is “public’s radiation dose”? The reviewer thinks the wording is too rough to be included in the summary.
Response 1: Thank you for your valuable feedback. We change the public’s radiation dose to the family members' radiation dose. [P.1, line 22]
Comments 2. On line 53, during BT, the seed is inserted into the prostate, not directly into the tumor. It may be difficult to identify the exact lesion on imaging.
Response 2: Thank you for your insightful comment. In prostate seeds implant is directly into the prostate. [P.2, line 53]
Comments 3. On line 53, a citation regarding this statement should be provided.
Response 3: Thank you for your insightful comment. We provide a citation regarding this statement. 6. Skowronek, J. (2017). Current status of brachytherapy in cancer treatment–short overview. Journal of contemporary brachytherapy, 9(6), 581-589. [P.2, line 53]
Comments 4. On line 54, what is “normal tissue”? surrounding tissues of the prostate?
Response 4: Thank you for your insightful comment. We change the “normal tissue” to protect “organs at risk (rectum, bladder)” [P.2, line 54]
Comments 5. Limitation is described incorrectly.
Response 5: Thank you for your insightful comment. This study limitation is sources air kerma strength differences, calibration tools uncertainty, verification sample size. We revise the limitation is described. [P.10-11, line 288-313]
Comments 6. Discussion is too short if it is only a discussion of the results of the experiment.
Response 6: Thank you for your insightful comment. We revise the discussion why this predictive model is relevant for clinical practice. Safety measures for the patient's family or others depend largely on the outcome of the predictive model. [P.10, line 242-282]
Comments 7. The authors should consider how it affects the actual treatment, how effective the treatment is, and how to minimize the impact on the surroundings and maximize the therapeutic effect.
Response 7: Thank you for your insightful comment. For the actual treatment protocol, we base it on American Brachytherapy Society (ABS) guidelines and European Society for Radiotherapy & Oncology (ESTRO) guidelines for effective treatment. We can use the volume study (preplan) to evaluate the total implant activity in the prostate. We used the ruler to measure the patient's AP length and RL length to calculate the external dose rate. We can use the predictive model to suggest minimizing radiation exposure and maximizing the therapeutic effect. We revise the statements about the minimize the impact on the surroundings.

Round 2
Reviewer 1 Report
Comments and Suggestions for Authors
Thank you for considering my comments on version 1 of the paper, which improved the paper. I still have comments on some phrases which are not clear and should be improved.
I also think that the ROC analysis with AUC figures is not suitable for the subject of this paper. I recommend to delete this ROC analysis from the paper.
See uploaded pdf with comments.

Comments on the Quality of English LanguageAuthors improved the paper. I still have comments on some phrases which are not clear and should be improved, see uploaded pdf
Author Response
Response to reviewer 1 comments
I would like to extend my sincere gratitude for the time and effort you have dedicated to reviewing our manuscript. Please find the detailed responses below and the corresponding revisions in the re-submitted files.
Point-by-point response to comments and suggestions for authors
Comments 1. The nine measurements were all normalized at 1 cm from the I-125 source in the water phantom.
Response 1: Thank you for your valuable feedback. We rewrite “The nine measurements were all normalized at 1 cm from the I-125 source in the water phantom.” to “The nine measurements were normalized to the measurement with the I-125 source at 1 cm in the water phantom.” [P.3, line 98-99]
Comments 2. long-term measurement of radiation
Response 2: Thank you for your valuable feedback. We rewrite “long-term radiation” to “long-term measurement of radiation” [P.3, line 101]
Comments 3. "normalized I-125 sources at 1 cm" is unclear! maybe authors want to say:
"normalized results to the result with the I-125 source at 1 cm in a water phantom"
Response 3: Thank you for your valuable feedback. We rewrite “normalized I-125 sources at 1 cm” to “normalized results to the result with the I-125 source at 1 cm in a water phantom” [P.3, line 112]
Comments 4. please rephrase; activity was xx cm? activity is not given in cm;
Response 4: Thank you for your valuable feedback. We rewrite to “The distance of the implant I-125 radioactive sources center to body surface was 10.33±1.42 cm for each patient in the anterior WET direction, 10.41±0.92 cm in the posterior WET direction, 17.19±1.51 cm in the left WET direction, and 17.24±1.53 cm in the right WET direction.” [P.4, line 149-152]
Comments 5. please skip the ROC analysis; to my opinions not suitable for this subject and it makes the paper more complicated than necessary.
Response 5: Thank you for your valuable feedback. We deleted all about ROC and AUC. [P.1, line 35-37] [P.4, line 156-158, line 164-165] [P.6, line 218-227] [P.10, line 268-275]
Comments 6. what are these differences? not the same as in sentences above? please make clear [P.5, line 182-190]
Response 6: Thank you for your valuable feedback. On page 5 lines 182 to 185 statements are only experimental measurement average difference at 0 cm, 30 cm and 100 cm. On page 5 lines 185 to 190 statements compare the experimental measurement and Monte Carlo simulation average difference at 0 cm,30 cm and 100 cm. [P.5, line 182-190]
Comments 7. as in my comments on version 1: explain what is no.=1, iy=1, mset =1
Response 7: Thank you for your valuable feedback.
no.=1: This parameter typically refers to a specific setting or configuration number within a simulation. It is commonly used in PHITS to denote the use of the first set of parameters or the first scenario being tested.
iy=1: This parameter refers to the selection of a specific mesh point or axis for analysis or output. In the context of a simulation involving a mesh grid, iy=1 indicates that the output or results are being extracted or visualized along the plane or line where y equals the first index, helping to display a two-dimensional dose distribution.
mset=1: PHITS includes some built-in coefficients in the [multiplier] section, which can tally dose. In the [multiplier] section, we must use "mset=1" to tally the ambient dose equivalent. [P.5, line 178, Figure 2.]
Comments 8. rephrase; suggestion: "normalized to the result with the I-125 source at 1 cm from the phantom surface"
Response 8: Thank you for your valuable feedback. We rewrite “normalized to the water phantom surface of 1 cm.” to “normalized to the result with the I-125 source at 1 cm from the phantom surface.” [P.7, line 232-233]
Comments 9. rephrase; unclear [P.9, line 238-242]
Response 9: Thank you for your valuable feedback. We rewrite this sentence “In validating the prediction model based on clinical measurement data, the calculation of H*(10) in the anterior and posterior direction of the patient's body surface (0 cm) showed a large difference in H*(10). The reason is that the prostate has a certain geometric volume, and there is a minor error in WET measurement (1-2 mm). The measured H*(10) is therefore distributed over a wide range and is significantly lower than the predictive model value by approximately 10%. In the calculation of H*(10) in the left and right directions of the patient's body surface (0 cm), the difference in H*(10) is small. When the radiation source insertion depth is greater than 15 cm, the difference in H*(10) is less than 5%.” [P.9, line 238-246]
Comments 10. incorrect English: a distribution of sources cannot be equal to H*(10); rephrase [P.9, line 248-250] and rephrase; not clear what authors want to state [P.9, line 251-252] rephrase; not clear what authors want to state [P.9, line 253-2255]
Response 10: Thank you for your valuable feedback. We rewrite “In practice, at distances greater than three times the size of the prostate, the ambient exposure dose rate H*(10) can be considered equal to the geometric center of gravity located at the implant, whose activity is equal to the total activity of the implant. The half-value layer of radiation emitted by I-125 seeds is approximately 2 cm WET for practical radiation protection purposes. This ambient exposure dose rate H*(10) value can be estimated based on the inverse square law and WET attenuation. Therefore, the H*(10) deviation will be reduced at distances 30 cm and 100 cm [15]”. [P.9, line 251-257]
Comments 11. 1 mGy
Response 11: Thank you for your valuable feedback. We rewrite the 1 mGy to 1 mSv. [P.9, line 258]

Reviewer 2 Report
Comments and Suggestions for Authors
none.
Author Response
I would like to extend my sincere gratitude for the time and effort you have dedicated to reviewing our manuscript. Please find the detailed responses below and the corresponding revisions in the re-submitted files.